# Learning Private Representations with Focal Entropy

## Abstract

How can we learn a representation with good predictive power while preserving user privacy? We present an adversarial representation learning method to sanitize sensitive content from the representation in an adversarial fashion. Specifically, we propose *focal entropy* - a variant of entropy embedded in an adversarial representation learning setting to leverage privacy sanitization. *Focal entropy* enforces maximum uncertainty in terms of confusion on the subset of privacy-related similar classes, separated from the dissimilar ones. As such, our proposed sanitization method yields deep sanitization of private features yet is conceptually simple and empirically powerful. We showcase feasibility in terms of classification of facial attributes and identity on the CelebA dataset as well as CIFAR-100. The results suggest that private components can be removed reliably.

## 1 Introduction

Lately, the topics of privacy and security are enjoying increased interest in the machine learning community. This can largely be attributed to the success of big data in conjunction with deep learning and the urge to create and process ever-larger data sets for mining. However, with the emergence of more and more machine learning services becoming part of our daily lives and making use of our data, special measures must be taken to protect privacy and decrease the risk of privacy creep Narayanan & Shmatikov (2006); Backstrom et al. (2007). Simultaneously, growing privacy concerns entail the risk of becoming a major deterrent in the widespread adoption of machine learning and the attainment of their concomitant benefits. Therefore, reliable and accurate privacy-preserving methodologies are needed, which is why the topic lately has enjoyed increased attention in the research community.

Several efforts have been made in machine learning to develop algorithms that preserve user privacy while achieving reasonable predictive power. Solutions proposed for privacy in the research community are versatile. A standard approach to address privacy issues in the client-server setup is to anonymize the *data* of clients. This is often achieved by directly obfuscating the private part(s) of the data and/or adding random noise to raw data. Consequently, the noise level controls the trade-off between predictive quality and user privacy (e.g., data-level Differential Privacy Dwork (2006)). These approaches associate a privacy budget with all operations on the dataset. However, complex training procedures run the risk of exhausting the budget before convergence. A recent solution to such a problem has been federated learning McMahan et al. (2016); Geyer et al. (2017), which allows us to collaboratively train a centralized model while keeping the training data decentralized. The idea behind this strategy is that clients transfer the *parameters* of the training model in the form of gradient updates to a server instead of the data itself. While such an approach is appealing to train a network with data hosted on different clients, transferring the models between clients and server, and averaging the gradients across the clients generates significant data transmission and extra computations, which considerably prolongs training. Another widely adopted solution is to rely on encoded data *representation*. Following this notion, instead of transferring the client's data, a feature representation is learned on the clients' side and transferred to the server. Unfortunately, the learned features may still contain rich information, which can breach user privacy Osia et al. (2017; 2018). Also, the extracted features can be exploited by an attacker to infer private attributes Salem et al. (2019). Yet, another approach is homomorphic encryption Armknecht et al. (2015). Despite providing strong cryptographic guarantees, in theory, it incurs considerable computational overhead, which still prevents its applicability for SOTA deep learning architectures Srivastava et al. (2019).

The recent success of adversarial learning in making the representations fair Louizos et al. (2015), unbiased Madras et al. (2018), and controllably invariant to sensitive attributes Xie et al. (2017), has led to the increased adoption of Adversarial Representation Learning (ARL) to control the private information encapsulated within the representation Roy & Boddeti (2019); Sadeghi et al. (2019). In the common ARL formalization of the privacy-preserving representation learning, a "predictor" seeks to extract the desired target attributes while an "adversary" seeks to reveal the private attributes. However, the solutions mentioned earlier can only meet its practical promises when the private attributes do not strongly correlate with the target attributes Roy & Boddeti (2019). In this paper, we deal with adversarial privacy-preserving representation learning. In this setting, the sensitive and target attributes are related to each other (e.g., 'Queen Elizabeth II.' and 'wearing hat', or 'Mahatma Gandhi' and 'wearing eyeglasses') to a large extent. The objective of this task is to learn a representation that contains all the information about non-sensitive attributes. At the same time, it omits to encode the sensitive attributes of them. Such representation can be transmitted to the server without concerns regarding the privacy revelation of classifiers having equal and higher capacity than the adversarial proxy used during training. For that, we adopt an ARL procedure and propose to learn a representation which maximizes the likelihood of the target information (i.e., *attribute predictor*) while increasing the uncertainty about the class that each sample belongs to (i.e., *class adversary*). With that, we intuitively tie the privacy notion to the class-level information and sanitize the class-revealing information from the representation in a semantic-aware fashion.

Specifically, we propose to learn the representation using the popular Variational Autoencoders (VAE) Kingma & Welling (2013), where the latent representation is additionally decomposed into two latent factors: *target* and *residual*. Whereas the target part encodes the information for the target task, the residual part identifies and collects the data's private part. In order to sanitize the target representation, we leverage an ARL procedure. There are two general strategies for ARL: the common solution for adversarial optimization is to maximize the loss of the adversary by minimizing the negative log-likelihood of sensitive variables. However, this is practically sub-optimal from the perspective of preventing information leakage. If the optimization does not reach the equilibrium, the resulting distribution associated with the minimum likelihood solution is subject to leaking the most amount of information. Another solution for adversarial optimization is to maximize the adversary's entropy by enforcing a uniform distribution over the sensitive labels Roy & Boddeti (2019); Sarhan et al. (2020). Such a solution provides no information to the adversary. However, it has the risk of weakening the encoder as it partially eliminates the adversary's role in the representation learning phase and is provably bound to the adversary's optimality. However, fulfilling the necessary optimality conditions impractical. Hence we seek to relax optimality by leveraging a quasi-optimal objective. To this end, we propose to maximize a variant of entropy - *focal entropy* - for dealing with inter-class uncertainty maximization. Focal entropy enforces the uncertainty to focus on a sparse set of similar classes and prevents the vast number of dissimilar classes from overwhelming the uncertainty. Maximization of focal entropy increases the uncertainty in a more organic, namely in a systematic and semantic-aware fashion. Hence, it is leading to a deeper privacy sanitization during the representation learning phase.

In summary, the main contributions of this paper are three-fold. **First**, we propose to learn the privacy-preserving representations. **Second**, we introduce an ARL setting for this task by adding a novel entropy term to the VAE. **Third**, we demonstrate experimentally that our proposed method learns a semantically meaningful privacy-preserving sanitized representation.

## 2 RELATED WORKS

Much research has been conducted in protecting differential privacy Dwork et al. (2017); Dwork (2006); Ryoo et al. (2017); Abadi et al. (2016) on *data* and *parameter* level by anonymizing raw data directly, or incorporating a randomized mechanism into the learning process, respectively. Although successful, our method is fundamentally different from them, as we aim to learn a private representation instead of preserving privacy in *data* or *parameter* level. While we do not consider their framework here, our method could employ differential privacy during the post-classifier training.

The advantages of learning and transmitting *representations* instead of data have been investigated recently in many works, see Osia et al. (2017; 2018), and references therein. Nevertheless, such a representation is proven to contain some privacy revealing information of clients. The recent

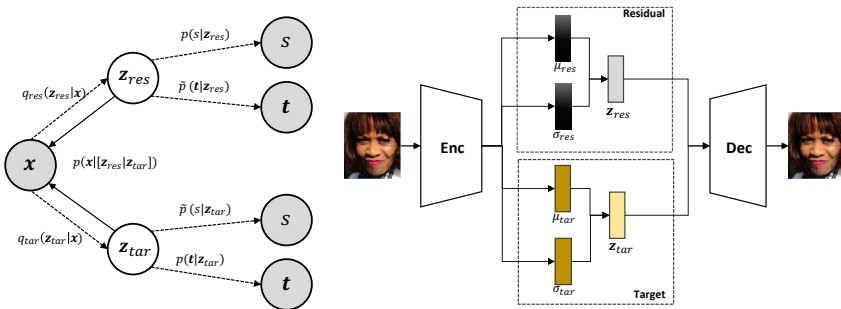

Figure 1: Schematic illustration of the proposed approach. **Left:** The graphical model associated with the minimax game. **Right:** Our proposed architecture with the two stream network is based on VAE, and augmented with additional predictor loss and (focal) entropy.

success of adversarial learning has led to the increased adoption of this technique for learning representations that preserve sensitive information in different types of data. For instance, Srivastava et al. (2019) proposed to learn privacy-preserving representations for automatic speech recognition (ASR). In Yang et al. (2018), a representation is learned on the raw student clickstream event data, captured as they watch lecture videos in massive open online courses. In Li et al. (2019), the authors proposed an obfuscator designed to hide privacy-related sensitive information from the features using adversarial training. Similarly, Kim et al. (2019) is based on adversarial learning, which encodes images to obfuscate the patient identity while preserving enough information for a medical segmentation task. Pittaluga et al. (2019) considered a formulation based on adversarial optimization between the encoding function and estimators for private tasks. Although our method is also based on adversarial learning, we differently facilitate adversarial sanitization using entropy. This leads to a more privacy-preserving representation while maintaining the method complexity.

The most related work to ours is by Roy & Boddeti (2019); Sadeghi et al. (2019), which aims at obtaining a sanitized representation using entropy and adversarial representation learning, respectively. Another related work to our paper proposed by Feutry et al. (2018) aims at learning representations that preserve the relevant part of the information while dismissing information about the private labels corresponding to the clients' identity. A key difference compared to this method is that they require labels for the downstream task during representation learning. Recently, Chen et al. (2018) proposed a complex method for privacy-preserving representation learning. Gabbay & Hoshen (2020) propose an approach for disentanglement using shared latent optimization and an asymmetric regularization. Edwards & Storkey (2016) propose sanitize representations utilizing an adversary. Liao et al. (2019a;b) employ an adversary to obtain a sanitized representation, however, also incorporating fairness constraints. Liu et al. (2018) proposed to use conventional entropy as an adversary in the context of a vanilla auto-encoder for privacy sanitization. To the best of our knowledge, our paper is the first work that proposes taking the class similarity into account for the adversary's entropy. Furthermore, fairness as proposed in Creager et al. (2019); Locatello et al. (2019); Quadrianto et al. (2019); Sarhan et al. (2020) is yet another intimately connected notion to privacy. While we do not consider fairness here, our method could also be extended to that problem, leading to exciting future research. Finally, we note that our model is different from the federated learning methodology in McMahan et al. (2016); Geyer et al. (2017), which focuses on learning a decentralized private model by sharing gradient updates instead of learning representations.

## 3 METHOD

We consider the scenario where we want to learn a dual-objective feature representation. The first objective entails providing high accuracy in terms of target attribute classification. The second objective entails minimizing information leakage w.r.t. sensitive attributes, ideally making classification impossible. To this end, we learn an encoder that is tasked to disentangle the attributes by decomposing the representation into two parts: target and residual partition. The target partition allows classification for the target attributes and is sanitized w.r.t. the sensitive attributes. In contrast,

the residual partition captures the information w.r.t. the sensitive attribute, subsuming the excess information. To drive the representation learning, we employ a Variational Auto-Encoder (VAE) Kingma & Welling (2013). Isolating information about sensitive and non-sensitive attributes to separate subspaces then corresponds to integrating the VAE into a minimax game between the target classifiers and the adversary. Leveraging VAE, in this regard, is beneficial in multiple respects. On the one hand, the VAE backbone allows us to learn better representations in particular when disentangling factors than employing solely a classical supervised approach Le et al. (2018); Gyawali et al. (2019). The representation learning objective takes advantage of the modeling flexibility and the large solution space of the VAE. With the objectives primarily orthogonal to each other, they act together rather than conflict. On the other hand, VAE allows for the integration of a notion of interpretability Charte et al. (2020); Ding et al. (2020). Integrating input reconstruction as a light-weight task in a minimax game facilitates understanding the sanitization process in a human-understandable visual way. This is a crucial point yet often overlooked in privacy.

**Background:** The representation learning problem is formulated as a game among *six* players: an encoder $\mathbf{E}$, a decoder $\mathbf{D}$, a target predictor $\mathbf{T}$, a sensitive attribute predictor $\mathbf{S}$, two adversarial classifiers $\tilde{\mathbf{T}}$ and $\tilde{\mathbf{S}}$. Whereas the non-adversarial predictors enforce the utility and the presence of the associated information in the representation, the adversarial predictors inhibit the undesirable information leakage and are directly responsible for disentanglement and sanitization of the representation. What is more, as the adversarial classifiers are learned during training, they act as surrogate adversaries for unknown oracle post-classifiers. The setup involves observational input $\boldsymbol{x} \in \mathcal{X}$, $M$ target attributes $\mathbf{t} \in \mathcal{A}_T = A_1 \times A_2 ... A_M$, each having $m_t$ classes $A_t = \{a_1, ..., a_{m_t}\}$, and a sensitive attribute $s \in \mathcal{A}_S = \{1, ..., N\}$ of $N$ classes.

The goal is to yield data representations from an encoder subject to privacy constraints. To learn the representation, a VAE is learned jointly along with several predictors. VAEs naturally decompose into two components – an encoder and a decoder.

The encoder $\boldsymbol{z} \sim q(\boldsymbol{z}|\boldsymbol{x}; \theta_E) : \mathcal{X} \to \mathcal{Z}$, parameterized by $\theta_E$ produces an embedding $\boldsymbol{z}$ in the latent space $\mathcal{Z}$. For the sake of privacy, we employ a modfied VAE encoder. Specifically, we propose an encoder that splits the generated representation into two parts: $\boldsymbol{z}_{tar} \in \mathcal{Z}_1$ represents the target part and residual part $\boldsymbol{z} \in \mathcal{Z}_2$. Concatenating the target and the residual parts $\boldsymbol{z} = [\boldsymbol{z}_{tar}|\boldsymbol{z}_{res}] \in \mathcal{Z}_1 \times \mathcal{Z}_2$ yields again the complete representation. For the following, we assume w.l.o.g. the target and the residual part of being of equal dimensionality. Encoding of the representation entails a stack of shared initial layers that diverge into parallel stacks from the penultimate layer onward into two separate network streams for disentangling the information. To this end, we let $\boldsymbol{z}_{tar} \sim q_{tar}(\boldsymbol{z}_{tar}|\boldsymbol{x}; \theta_E^{tar}) : \mathcal{X} \to \mathcal{Z}_1$ denote the encoder of the target representation stream and the encoder of the $\boldsymbol{z}_{res} \sim q_{res}(\boldsymbol{z}_{res}|\boldsymbol{x}; \theta_E^{res}) : \mathcal{X} \to \mathcal{Z}_2$ the residual stream, respectively. Given the two encoding streams (with partially shared parameters), $\theta_E^{tar} \subset \theta_E$ and $\theta_E^{res} \subset \theta_E$ denoting the parameters of the networks stream. Given the *common parameters* in the *shared stack*, $\theta_E^{tar} \cap \theta_E^{res} \neq \emptyset$. The second component in the VAE, the decoder is given by $\boldsymbol{x} \sim p(\boldsymbol{x}|\boldsymbol{z}; \theta_D) : \mathcal{Z} \to \mathcal{X}$, which seeks to reconstruct the observational input, and is parameterized by $\theta_D$.

Next, we define the predictors for the target and sensitive attributes. First, the target predictor is given as $p_T(\mathbf{t}|\boldsymbol{z}_{tar}; \theta_{tar}) : \mathcal{Z}_1 \to \mathcal{A}_T$. Second, the sensitive attribute predictors is given as $p_S(s|\boldsymbol{z}_{res}; \theta_{res}) : \mathcal{Z}_2 \to \mathcal{A}_S$, parameterized by $\theta_{tar}$ and $\theta_{res}$. Last, we have the associated adversarial predictors denoted as $\tilde{p}_T(\mathbf{t}|\boldsymbol{z}_{res}; \tilde{\theta}_{res}) : \mathcal{Z}_2 \to \mathcal{A}_T$ and $\tilde{p}_S(s|\boldsymbol{z}_{tar}; \tilde{\theta}_{tar}) : \mathcal{Z}_1 \to \mathcal{A}_S$, parameterized by $\tilde{\theta}_{tar}$ and $\tilde{\theta}_{res}$. Note, that the difference between predictors and their adversary is their swapped input source.

**Optimization:** Learning the representation formalizes as the optimization of multi-player nonzero-sum game given as:

$$\min_{\theta_{E,T,D,S}} \max_{\theta_{\tilde{T},\tilde{S}}} \lambda \cdot \phi_{ED}(\theta_E, \theta_D) + \alpha_T \cdot \phi_T(\theta_E, \theta_{tar}) +$$

$$\alpha_S \cdot \phi_S(\theta_E, \theta_{res}) + \beta_{\tilde{T}} \cdot \phi_{\tilde{T}}(\theta_E, \tilde{\theta}_{res}) + \beta_{\tilde{s}} \cdot \phi_{\tilde{S}}(\theta_E, \tilde{\theta}_{tar}), \tag{1}$$

with $\phi_*(.|.)$ denoting the players, and $\lambda, \alpha_*, \beta_* \in \mathbb{R}$ weighting scalars. The hyperparameters allow for a trade-off between the utility of target classification and the latent code's privacy preservation. The associated process schematic is shown in Fig. 1. As the predictors and the adversarial predictors have a competitive relationship, they are optimized differently. Whereas predictors are trained using cross-entropy minimization, the adversarial predictors seek to maximize entropy, which will be

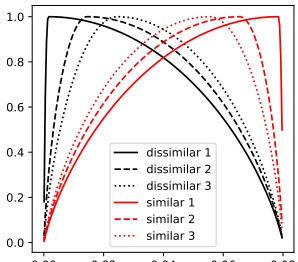 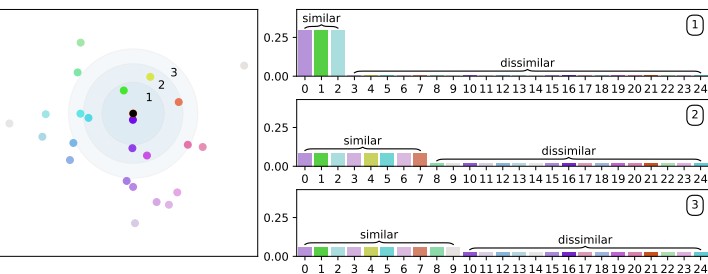

Figure 2: Illustration of focal entropy and the effect of grouping assuming equal partitioning of probability mass. **Center:** Visualization of a sample configuration; schematic focus regions depicted as circles ranging from narrow (1) to wide (3). **Left:** Visualization of off-center entropies (similar, dissimilar) for different focus regions scenarios. The more narrow the focus, the more weight "*similar*" samples have. Conversely, the wider focus range, the more equiprobability is approached. **Right:** Entropy visualization for focus scenarios.

explained in the forthcoming sections. The latent space should be disentangled at the equilibrium of this minimax game, yielding the representation with the desired privacy sanitization properties.

### 3.1 Adversarial Representation Learning

In the following, we will define the optimization criterion for each player. First, we explain the predictors, followed by the reconstruction, and conclude with the adversarial classifiers' optimization, notably sanitization and herein focal entropy.

**Reconstruction:** Leveraging VAE as backbone, optimization of $\phi_{ED}$ entails minimization of Evidence Lower Bound (ELBO). The ELBO is defined as:

$$\phi_{ED}(\theta_E, \theta_D) = \mathrm{E}_{q(\boldsymbol{z}|\boldsymbol{x};\theta_E)} \left[ \log p\left(\boldsymbol{x}|\boldsymbol{z};\theta_D\right) \right] - D_{KL}(q\left(\boldsymbol{z}|\boldsymbol{x};\theta_E\right) \| p\left(\boldsymbol{z}\right)), \tag{2}$$

which decomposes into two parts. The first term corresponds to the reconstruction likelihood measuring the error in input reconstruction. The second term corresponds to the prior constraint, where we assume isotropic Gaussian as latent prior $p\left(\boldsymbol{z}\right) = \mathcal{N}\left(0, \boldsymbol{I}\right)$, with $\boldsymbol{I}$ denoting the identity matrix. Furthermore, we assume the posterior approximates to correspond multivariate Gaussians. Given two *separate* encoding streams, one for the target and the residual part, we yield: $q_{tar}\left(\boldsymbol{z}_{tar}|\boldsymbol{x};\theta_E^{tar}\right) = \mathcal{N}\left(\boldsymbol{\mu}_{tar}, \mathrm{diag}(\boldsymbol{\sigma}_{tar})\right)$ and $q_{res}\left(\boldsymbol{z}_{res}|\boldsymbol{x};\theta_E^{res}\right) = \mathcal{N}\left(\boldsymbol{\mu}_{res}, \mathrm{diag}(\boldsymbol{\sigma}_{res})\right)$, with $\boldsymbol{\mu} \in \mathbb{R}^h$ and $\mathrm{diag}(\boldsymbol{\sigma}) \in \mathbb{R}^{h \times h}$ the diagonal matrix constructed from vector $\boldsymbol{\sigma} \in \mathbb{R}^h$.

**Predictors:** The predictors are all modeled as conditional distributions. They are trained by:

$$\phi_T(\theta_E, \theta_{tar}) = D_{KL}(p(\mathbf{t}|\boldsymbol{x}) \| p_T(\mathbf{t}|\boldsymbol{z}_{tar}); \theta_{tar}) \tag{3}$$
$$\phi_S(\theta_E, \theta_{res}) = D_{KL}(p(s|\boldsymbol{x}) \| p_S(s|\boldsymbol{z}_{res}); \theta_{res}) \tag{4}$$

where $p(\mathbf{t}|\boldsymbol{x})$ and $p(s|\boldsymbol{x})$ denote the ground-truth labels for training input $x$ for the target attribute and the sensitive attribute, respectively. $D_{KL}(.\|.)$ denotes the Kullback-Leibler divergence.

**Sanitization:** In order to minimize the information leakage across representation partitions, we leverage the maximization of entropy. This enforces the representation to be maximally ignorant w.r.t. attributes in the domain confusion sense. Specifically, for the adversarial target attribute, we let

$$\phi_{\tilde{T}}(\theta_E, \tilde{\theta}_{res}) = D_{KL}(\tilde{p}_T(\mathbf{t}|\boldsymbol{z}_{res}) \| \mathcal{U}; \tilde{\theta}_{res}), \tag{5}$$

where $\mathcal{U}$ denotes the uniform distribution. Although maximization of entropy is sufficient for non-private attributes to minimize information leakage across representation partitions, we postulate that proper sanitization must be conducted w.r.t. to focus classes in a similarity-aware fashion. As such, we aim to sanitize the information of each observation within a focus area, i.e., nearest neighbors (NN). For each observation, the NNs share the most commonalities in features. As such, discrimination w.r.t. NNs is less likely to be trivially achievable. To this end, training of $\phi_{\tilde{S}}(\theta_E, \tilde{\theta}_{tar})$ leverages a modification of entropy for deep sanitization.

## 3.2 FOCAL ENTROPY

Focal entropy aims at deep sanitization of sensitive information by leveraging a notion of similarity in terms of the sensitive attribute. To this end, it requires partitioning the sensitive attribute into two sets w.r.t. mutual discriminativeness: "*similar*" classes (A) "*dissimilar*" classes (B). This partitioning is input specific and is conducted either according to **a)** label information, or **b)** using some scoring function obtained using a pre-trained model or "*on-the-fly*" during training. Thus, let $\mathbf{r}(\boldsymbol{x})$ be the scores given the observational input $\boldsymbol{x}$, which we will simply denote as $\mathbf{r}$ for the sake of economy of notation. Then the class predictions given are by the index with maximal score, i.e., $p(y|x; \boldsymbol{\theta}) = p(\mathbf{r}) \in \arg\max \mathbf{r}$. Then "*similar*" consists of the set of labels corresponding to the $k$-largest scores, and "*dissimilar*" the complement:

$$\mathcal{A}_{Similar} = \left\{ s_i \in \mathcal{A}_S^{(k)} : \forall i \in \{1, ..., k\}, r_{s_i} \geq r_{[k]} \right\}, \quad \mathcal{A}_{Dissimilar} = \mathcal{A}_S \setminus \mathcal{A}_{Similar} \qquad (6)$$

Here $r_{[k]}$ denotes the $k$-th largest element of $\mathbf{r}$, and $\mathcal{A}_S^{(k)}$ denotes the set of $k$ distinct elements in $\mathcal{A}_S$. Furthermore, we let $N_A, N_B$ denote to the number of classes in $A$ and $B$, where typically $N_A \ll N_B$ holds and $N = N_A + N_B$.

Given the just defined notion of similarity, focal entropy aims at establishing uniformity in terms of likelihood (w.r.t. the sensitive attribute) within each group – with the probability mass divided *equally* between the two groups (A) and (B). Given $N_A \neq N_B$, this implies reciprocal re-weighting of the classes in $A$ and $B$, respectively.

Assuming $N_A \ll N_B$, this aims at giving proportionally more weight to confusion w.r.t. members of similar classes. Analogously, members of dissimilar classes are down-weighted accordingly. Consequently, the classifier is forced to be maximally ignorant w.r.t. class properties of the similar class members that share high correlation. The feature representation produced from the encoder should not have any sensitive information in the target part. As this approach implicitly is driving sanitization by focusing on specific targets, we refer to this as "*focal entropy*". Implementation of the focal entropy criterion is essentially equivalent to maximization w.r.t. an off-centered entropy Lallich et al. (2007) in the special case of normalized uniform probability within each group. That is, in contrast to conventional entropy that takes its maximal value when the distribution of the class variable is uniform $(\frac{1}{N}, ..., \frac{1}{N}) = \mathcal{U}$, focal-entropy takes its maximum non-centered. See Fig. 2 for the schematic illustration of the proposed concept.

Specifically, focal entropy seeks to have the entropy peak at $\boldsymbol{\tau} \in \mathbb{R}^N$, with $\boldsymbol{\tau} \neq \mathcal{U}$. Assuming equally divided probability mass, we yield $\boldsymbol{\tau}$ defined as:

$$\tau_{1,...,N_A} = \frac{N_B}{N_A^2 + N_B \cdot N_A}, \tau_{N_A+1,...,N} = \frac{N_A}{N_B^2 + N_A \cdot N_B}, \text{ s.t. } \sum_i^N \tau_i = 1. \qquad (7)$$

Further, let $h(.)$ denote the entropy w.r.t. a vector $\mathbf{p} = (p_1, ..., p_N) \in \mathbb{R}^N$ of probabilities with $\sum_i^N p_i = 1$ defined as $h(\mathbf{p}) = \sum_{j=1}^N p_j \log p_j$. In order to achieve entropy achieving its maximum at $\boldsymbol{\tau}$, the class probabilities $\mathbf{p}$ have to be transformed. To that end, each $p_i \in [0, 1]$ is mapped to a corresponding $\pi_j$, according to:

$$\pi_j = \frac{p_j}{N\tau_j} \quad \text{if } 0 \leq p_j \leq \tau_j, \quad \pi_j = \frac{N(p_j - \tau_j) + 1 - p_j}{N(1 - \tau_j)} \quad \text{if } \tau_j \leq p_j \leq 1. \qquad (8)$$

In order to fulfill the properties of an entropy, the $\pi_j$ have to be normalized subsequently. This is achieved according to $\pi_j \cdot (\sum_i^N \pi_i)^{-1} = \pi_j^*$. Thus we yield the off-centered entropy $\eta(.)$ of probabilities $\mathbf{p}$ defined as $\eta(\mathbf{p}) = h(\boldsymbol{\pi}^*) = -\sum_{j=1}^N \pi_j \log \pi_j$. Consequently, adversarial sanitization seeks to maximize the off-centered entropy w.r.t. private attribute on $\boldsymbol{z}_{tar}$ according to:

$$\phi_{\tilde{S}}(\theta_E, \tilde{\theta}_{tar}) = \eta(\sigma_{\boldsymbol{z}_{tar}}^{\tilde{S}}), \qquad (9)$$

with $\sigma_{\boldsymbol{z}_{tar}}^{\tilde{S}} \in \mathbb{R}^N$ denoting the softmax vector of the adversarial predictor $\tilde{p}_S(y|\boldsymbol{z}_{tar}; \tilde{\theta}_{tar})$.

Maximizing the off-centered entropy is analogous to minimizing the Kullback-Leibler divergence w.r.t. $\boldsymbol{\tau}$, such that we yield:

$$\phi_{\tilde{S}}(\theta_E, \tilde{\theta}_{tar}) = D_{KL}(\tilde{p}_T(s|\boldsymbol{z}_{tar}) \| \boldsymbol{\tau}; \tilde{\theta}_{tar}). \qquad (10)$$

| **CIFAR-100** Krizhevsky (2009) | | |
|---|---|---|
| Method | Tar. | Adv. |
| Random Chance | 1.0 | 0.01 |
| Roy & Boddeti (2019) | 0.71 | 0.16 |
| Sadeghi et al. (2019) | 0.80 | 0.16 |
| **Ours** | **0.82** | **0.16** |
| **CelebA** Guo et al. (2016) | | |
| Method | Tar. | Adv. |
| Random Chance | 1.0 | < 0.001 |
| Szabo et al. (2018) | - | 0.09 |
| Harsh Jha et al. (2018) | - | 0.14 |
| Denton & Birodkar (2017) | - | 0.03 |
| Gabbay & Hoshen (2020) | - | **< 0.01** |
| Bouchacourt et al. (2018) | 0.88 | 0.178 |
| Liu et al. (2015) | 0.873 | - |
| Zhang et al. (2014) | 0.854 | - |
| Liu et al. (2018) | 0.878 | - |
| **Ours** | **0.90** | **< 0.01** |

Table 1: Results on CelebA and CIFAR-100.

| **CIFAR-100** Krizhevsky (2009) | | |
|---|---|---|
| Method | Tar. | Adv. |
| Random Chance | 1.0 | 0.01 |
| Ours (`entropy`) | 0.70 | 0.16 |
| **Ours (`F. entropy`)** | **0.82** | **0.16** |
| **CelebA** Guo et al. (2016) | | |
| Method | Tar. | Adv. |
| Random Chance | 1.0 | < 0.001 |
| Ours (`entropy`) | 0.90 | 0.061 |
| **Ours (`F. entropy`)** | **0.90** | **< 0.01** |

Table 2: Ablation study (*Focal entropy*) on CIFAR-100 and CelebA.

| **CelebA** Guo et al. (2016) | | |
|---|---|---|
| Method | Tar. | Adv. |
| Random Chance | 1.0 | < 0.001 |
| Ours (`normal cls.`) | 0.90 | 0.007 |
| Ours (`strong cls.`) | 0.90 | 0.009 |

Table 3: Probing analysis on CelebA.

It should be noted that focal entropy contrasts with conventional entropy in terms of promoting hub formation. Here a hub relates to the adversarial mapping of multiple sensitive "similar" entities, i.e., IDs for CelebA, to the same "target". Essentially, entropy drives for equiprobability of classes and suppresses the formation of hubs, whereas focal entropy enforces the collapse of "*similar*" sensitive entities on the same hub. Hence, entropy induces a *bijective* mapping due to its invariance to random relabeling, whereas focal entropy induces a *surjective* mapping w.r.t. hubs. It should be noted that surjective mappings give rise to a *loss of information* as the input is not guaranteed to be recovered through inversion Nielsen et al. (2020).

## 4 EXPERIMENTS

**Setup:** We experimentally validate the proposed method using two datasets, which exhibit different characteristics. First, the algorithm is tested on CIFAR-100 Krizhevsky (2009) dataset. It consists of 100 classes, grouped into 20 superclasses. The classes are (theoretically) entirely mutually exclusive, such that each item is associated with precisely one fine-grained category belonging to one coarse superclass. In the privacy setting, the encoder is forced to learn a representation that is superclass aware while not revealing any information about the fine-grained classes. Second, the CelebA dataset of celebrity face images Guo et al. (2016). This dataset contains a large number of identities (people) with multiple observations of each. The "in-the-wild" nature of face images offers a richer testbed for our method as both identities, and contingent factors are significant sources of variation. Furthermore, the class structure does not permit a well-defined separation between public and private spaces.

**Implementation Details:** We employ Stochastic Gradient Descent (SGD) with momentum $0.9$ and weight-decay of $1e-4$ and a batch size of $200$, with a learning rate of $0.01$. Furthermore, we employ learning rate decay at every 30th epoch by a factor of $0.1$. On CelebA, to learn target and adversary classifiers smoothly, we perform pre-training without adversary objective for 100 epochs. This allows us to learn a stable representation crucial for gentle modification, i.e., disentangling the information of the sensitive and the target variable. Subsequently, the model is trained for another 1200 epochs, allowing for a slow "burn-in" of adversary objectives such it cannot be undone. On CIFAR-100, the model is trained for 3000 epochs. As for the trade-off parameters of Eq. 1, we employ the following scheme. CIFAR: equal weights for target loss and adversary: $\alpha_S = \alpha_T = 1$, $\beta_T = \beta_S = 1$, NN size of $5$. CelebA: weighting according to number of objectives $\alpha_T = \frac{1}{40}, \alpha_S = 1.0, \beta_T = \frac{1}{40}, \beta_S = 1.0$. For reconstructing weighting, we always assume $\lambda = 1.0$. For grouping in focal entropy, we assume the $k-$NN size of $k = 16$.

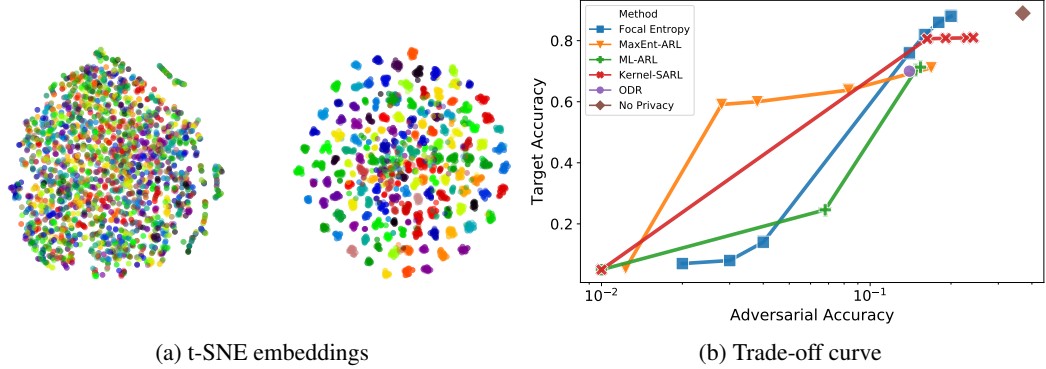

(a) t-SNE embeddings                    (b) Trade-off curve

Figure 3: **Left:** t-SNE embedding of latent representation for a subset of 200 IDs. **Left t-SNE:** embedding of the target part $z_{tar}$. **Righ t-SNE:** embedding of residual part $z_{res}$. t-SNE is unable to reveal any sort of regular structure in $z_{tar}$ w.r.t. private attribute. **Right:** Trade-Off curve between target accuracy and adversarial accuracy on CIFAR-100.

### 4.1 PRIVACY SANITIZATION ANALYSIS

We evaluated the privacy sanitization in terms of target and adversarial accuracy on two benchmarks:

**CIFAR-100:** As the first testbed, we adopt the "simulated" privacy problem proposed by Roy & Boddeti (2019) designed on the CIFAR-100 dataset. We treat the *coarse* (superclass) and *fine* (class) labels as the target and sensitive attribute, respectively. The task is to learn the superclasses' features while not revealing the information about the underlying classes. Here we note that "ideally" we desire a predictor accuracy of 100%, an adversary accuracy of 1% (random chance for 100 classes). In Tab. 1, we report the accuracy achieved by the attribute predictor and adversary. From these results, we observe that with our proposed method, the representation achieves the best target accuracy while being comparable on adversary accuracy compared to the SOTA sanitization methods Roy & Boddeti (2019); Sadeghi et al. (2019). Using conventional entropy for adversary instead of our proposed *focal entropy* results in a significant performance drop (from 82% to 70% in terms of target accuracy at comparable adv. accuracy). In Fig. 3b, we report the characteristics of the proposed approach in terms of trade-off curves that portray the correlation between privacy and target utility. We compared with the most relevant works to us: MaxEnt-ARL and ML-ARL from Roy & Boddeti (2019), Kernel-SARL Sadeghi et al. (2019), Orthogonal Disentangled Representations (ODR) from Sarhan et al. (2020), and vanilla baseline *(No Privacy)* without sanitization. As can be seen, the higher the correlation to privacy, the higher appears the loss of accuracy at a high level of sensitive accuracy. Furthermore, the proposed approach features a significantly better utility trade-off in the high target-accuracy domain. Additionally, to assess the effect of our proposed *focal entropy* in terms of the adversarial component, comparing it with conventional entropy. We report the results in Tab. 2.

**CelebA:** As the second testbed, we adopt the CelebA dataset, which has a richer structure to be utilized for privacy while being less structured in terms of similarity. Specifically, we treat the attribute labels as target and celebrity identity (i.e., ID) as the source of sensitive information. The task is to learn to classify the attributes while not revealing the information about the identities. We note that ideally, we desire a lower adversary accuracy ($< 0.001$) compared to the previous case, as the number of ID classes is an order of magnitude higher. In Tab. 1 we report the target/adversary accuracy. We observe that the representation we learn achieves a higher target accuracy and lower adversary accuracy than the strong baseline ML-VAE Bouchacourt et al. (2018). Using standard entropy instead of our proposed *focal entropy* in CelebA shows a considerable higher privacy leakage (namely, 0.061 and $< 0.01$) - see Tab. 2. Moreover, to analyze the sanitization of privacy in the latent representation, we visualize the t-SNE of the target and residual parts of the representation in Fig. 3a. For visualization, a subset of 200 IDs was chosen randomly from the test set. As can be seen, the private class associations appear random and cannot be recovered from the target part. This contrasts sharply with the residual part, where common IDs form clusters, giving rise to a proper margin, confirming our quantitative results in terms of target/adversary accuracy. Additionally, we

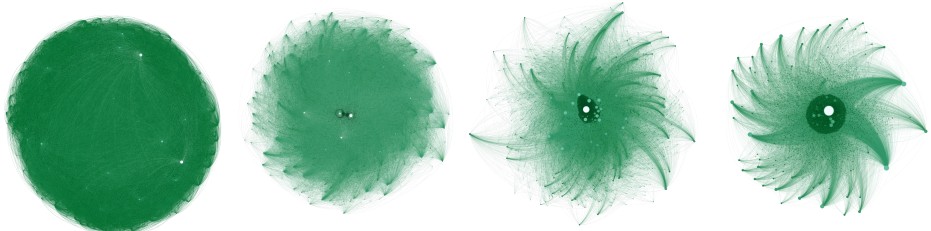

Figure 4: Visualization of adversary ID re-mapping graph on CelebA for entropy to focal entropy with different k-NNs on $z_{tar}$. Nodes correspond to IDs, edges to adversarial re-mapping of an ID to facilitate adversary confusion. Node size/brightness scales with number of associations (the bigger/brighter, the more IDs are mapped to a specific node). From *left* to *right* increasing $k$ for focal entropy: $k = 1$ ($\approx$ standard entropy), 2, 16 and 64.

performed a probing experiment to assess the classifier capacity's dependency regarding the learned representation's privacy leakage potential. Specifically, we determined the adversarial accuracy on the learned representation w.r.t. to the classifier's strength. The `strong` probing classifier has approximately double the proxy classifier's capacity that was used for training - see Tab. 3.

## 4.2 HUB ANALYSIS

This section provides an analysis of how the application of focal entropy, with its integration of the notion of $k-$NN, promotes the formation of "hubs". By varying the neighborhood size $k$, focal entropy manifests itself between two extremes: **i)** Choosing a *small* $k$, focal entropy in the limit ($k = 1$) approaches conventional entropy - no hubs are formed - remapping is a "*one-to-one*" correspondence. This can be attributed to conventional entropy being invariant to random relabeling, inducing a bijective mapping, giving rise to information preservation. This is prone to sub-optimal solutions since the forced equalization of classes by disregarding semantic similarity of classes makes the model susceptible to find shortcuts such as label swapping. In this regard, target accuracy oscillations co-occurring with degenerate ID-remappings such uniformity across all labels or temporary collapse to very few hubs were observed. **ii)** Choosing a *large* $k$, focal entropy promotes the formation of a single dominant hub, which is also referred to as the classical hubness problem. This phenomenon is related to the convergence of pairwise similarities between elements to a constant as the space's dimensionality increases Radovanović et al. (2010) (collapsing on a single hub/trivial solution). The single hub then manifests itself as the favored result of queries Dinu et al. (2014) - giving rise to trivial solutions rather than deep sanitization. Choosing a *non-extreme* $k$ leads to the formation of numerous equisized hubs, inducing a surjective mapping. As the formation of multi-hubs coincides with a collapse, less information is required to establish a mapping. Hence, this characteristic gives rise to information removal and is thus imperative for proper sanitization. Figure 4 depicts the graphs induced by remapping for four different neighborhood configurations. Analyzing the nodes' average degree in the corresponding graphs, we observe a continuous decrease with growing neighborhood size $k$. Specifically, starting with entropy and increasing the increasing $k$ for focal entropy, we yield the average degrees of: 13.91, 7.51, 7.2, and 3.0.

## 5 CONCLUSION

In this paper, an adversarial representation learning method is proposed to deal with a setting where the target and sensitive attributes are related to each other to a large extent. Training the representation entails decomposition into a target and a residual part. Here, the target part is shareable without privacy infringement and facilitates applicability for a target task. In contrast, the residual part subsumes all the private non-shareable information. Our proposed adversarial learning method employs focal entropy for deep sanitization of privacy from the representation. Subsequently, our experiments confirm that our proposed method learns sanitized data representations using a manageable level of supervision. Future work will entail adaptive and "*on-the-fly*" similarity grouping during training.

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

# Learning Private Representations with Focal Entropy (Appendix)

**Anonymous authors**

In the following sections, we add additional details omitted in the main paper due to space restrictions. In Sec. 1, we present an *ablation* study on different components of the loss function, underlining the importance of each term. In Sec. 2, we analyze the impact of varying the neighborhood size of $k-$NN in focal entropy on the adversarial accuracy with experiments conducted on the CelebA dataset. In Sec. 3, the sanitization convergence behavior of different classifiers involved in the adversarial minimax game is analyzed. In Sec. 4, we analyze the effect of the *classifier-strength* on the privacy leakage and dependence on training time. In Sec. 5, we present more visualizations around the concept of *hub formation* and the connection to focal entropy. It contains a visualization of hub forming identities and zoom-in visualization of the adversarial remapping of the identities in CelebA. Next, we generated reconstructions on CelebA dataset samples generated in Sec. 6. In Sec. 7, we present detailed results of the accuracy and privacy trade-off on *all* the attributes in CelebA dataset. Finally, *architectural details* are presented in Sec. 8.

## 1    Ablation Analysis on Loss Components

To assess the contribution of each component of our objective function, we evaluated each module's performance separately, gradually adding components: *reconstruction loss, target classification loss, adversary loss*. For that, we report the ablation study of the loss components on CelebA in Tab. 1. Furthermore, Fig. 1a shows the dependency between adversarial accuracy and the number of training epochs. As can be seen, beyond 1000 epochs, the adversarial accuracy drops below $< 0.01$, nearing chance level.

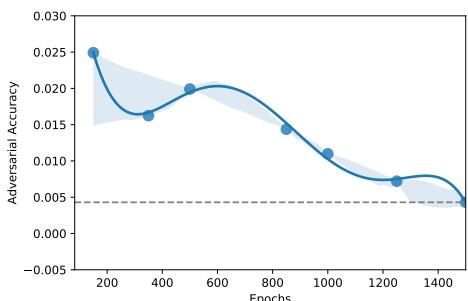

(a) Adversarial classifier training time dependency

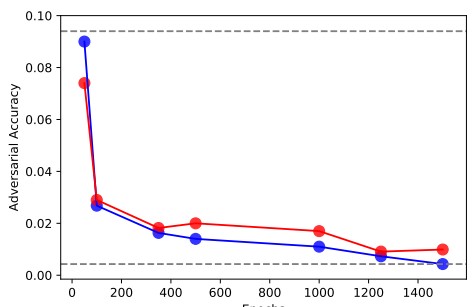

(b) Normal and strong adversarial classifier training time dependency

Figure 1: **Left:** Relationship between adversarial accuracy and the number of training epochs on CelebA. The translucent band corresponds to 50% confidence minimum and maximum adversarial accuracy, respectively. **Right:** Relationship between adversarial accuracy for strong (red) and normal classifier (blue) w.r.t. the number of training epochs on CelebA. The translucent band corresponds to 50% confidence interval. Dashed lines correspond to minimum and maximum adversarial accuracy, respectively.

## 2 NEIGHBORHOOD SIZE

Focal entropy entails a notion of similarity tied to the integration of $k-$NN. Here, we study the effect of varying $k$ on focal entropy and the associated the adversary accuracy. See Fig. 2 for a visualization of this relationship on the CelebA dataset. As can be seen, the adversary accuracy has oscillatory behavior with various local minima, reaching optimum around $k = 16$. This can be attributed to the superpositioning of different hubs, each exhibiting a different similarity pattern. Analysis of hub formation is explained in Sec. 4.2 in the main paper and Fig. 4 in the main paper.

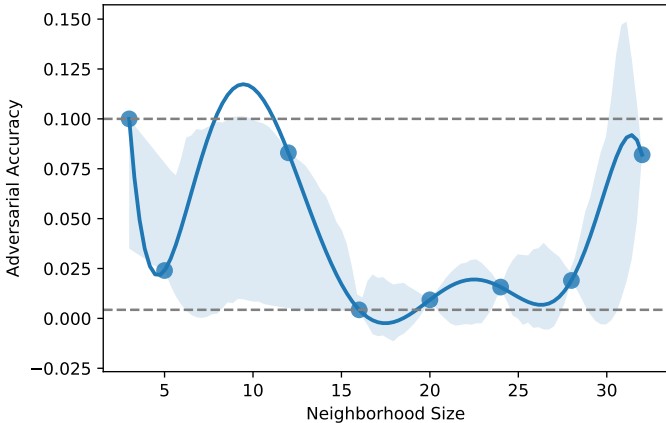

Figure 2: Relationship between adversary accuracy and $k-$NN size on CelebA dataset. Translucent band corresponds to 50% confidence interval. Dashed lines correspond to minimum, and maximum adversary accuracy, respectively.

## 3 SANITIZATION CONVERGENCE BEHAVIOUR

This section explores the behavior of standard entropy and the proposed focal entropy for sanitization. Fig. 3 depicts the classification performance during the training of different classifiers involved in the minimax optimization scheme: target classifier accuracy, adversarial sensitive attribute accuracy, and sensitive attribute accuracy. As can be seen, employing standard entropy for sanitization results in re-occurring patterns of oscillations. This can be attributed to degenerate/trivial solutions and "shortcuts". In contrast to that, focal entropy shows a relatively smooth convergence behavior. More details on the behavior can be found in Sec. 4.2 in the main paper.

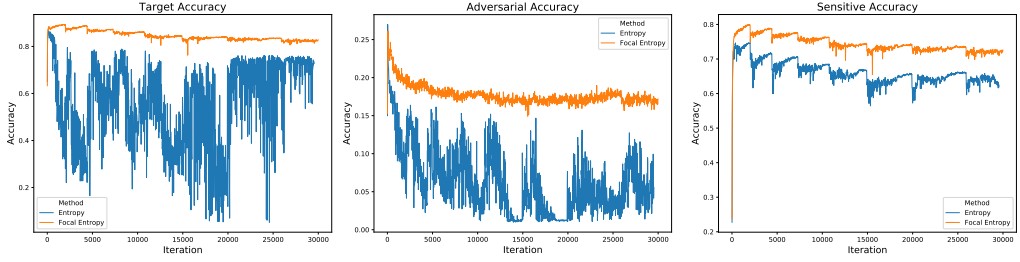

Figure 3: Sanitization convergence behavior of standard entropy and focal entropy on CIFAR-100 for different classifiers: **Left:** Target accuracy, **Center:** Adversarial accuracy, **Right:** Sensitive attribute accuracy

## 4 PROBING ANALYSIS WITH STRONG CLASSIFIER

This section provides more detail on assessing the classifier strength in terms of privacy leakage and the dependence on training time. We thereby largely follow the protocol of Harsh Jha et al. (2018); Sadeghi et al. (2019). Specifically, we employed a *stronger* post-classifier (Tab. 3 in the main paper) compared to the one used for learning the representation. The stronger post-classifier is endowed with additional layer stack (see Tab. 6c for architectural details), trained for 100 epochs. The results in Tab. 3 of the main paper suggest no significant changes in target and adversarial accuracy. Figure 1b extends these results, depicting the relationship w.r.t. the number of epochs. As can be seen, the difference between the *strong* and *normal* classifier is largely constant, independent of the epoch.

| | **CelebA** Guo et al. (2016) | |
|---|---|---|
| **Method** | **Target Acc.** | **Adversarial Acc.** |
| Upper-bound / Random Chance | 1.0 | < 0.001 |
| Our Method (only *Rec.* loss) | 0.88 | - |
| Our Method (*Rec. + Tar.* loss) | 0.91 | 0.751 |
| Our Method[full] (*Rec. + Tar. + Adv.* loss) | 0.90 | < 0.01 |

Table 1: Ablation analysis for loss components on CelebA dataset.

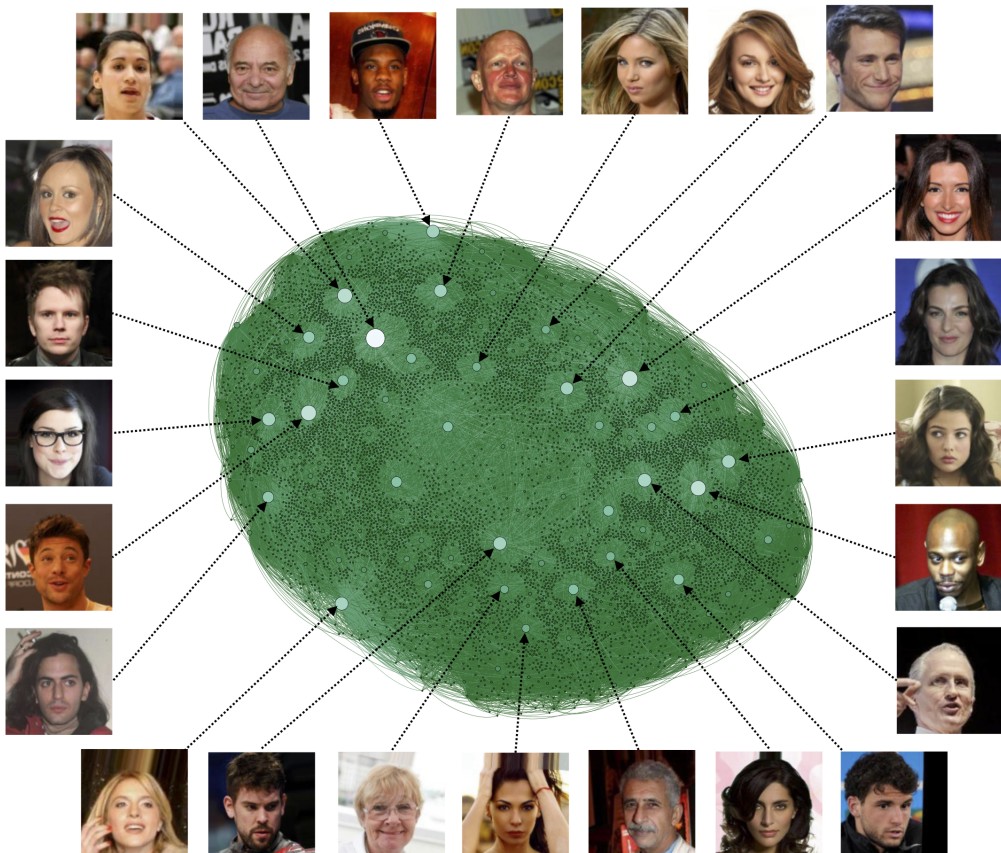

Figure 4: Visualization of CelebA identities of adversary classification network. The network (green) corresponds the $k-$nearest neighborhood size $k = 5$.

## 5 Extended Hub Analysis

This section provides a visually more detailed analysis of how the application of focal entropy promotes the formation of "hubs" (explained in Sec. 4.2 in the main paper). To study that, we analyzed the identity remapping of IDs on the CelebA dataset. Employing focal entropy results in a surjective ID confusion pattern by taking similar IDs into account for privacy sanitization.

### 5.1 Visualization of Hub Faces:

To study hubs' semantics, we visualize the CelebA identities of the network corresponding to focal entropy with $k-$nearest neighborhood size $k = 5$. See Fig. 4 for the visualization of the hub faces. As can be seen, the hubs exhibit a rich diversity in facial properties.

### 5.2 Adversarial Identity Mapping:

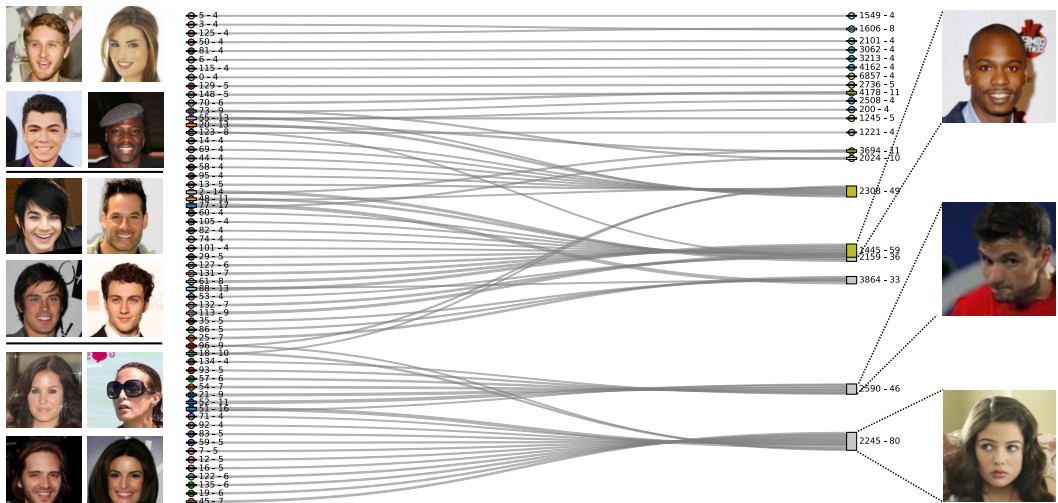

Figure 5: Visualization of the remapping of IDs in CelebA due to adversarial representation learning. Source IDs (left) are remapped to new target IDs (right). Pictures on the left are samples that get mapped to a hub; separation with bar indicates different target hub. Pictures on the right are the visualization of hub identities. Visualization contains a subset of 150 IDs, with targets getting at least four associations. The first number at each node indicates the ID, the second the number of images per ID. Node splicing indicates the remapping of a single ID to multiple adversarial targets.

Figure 5 is a zoom-in version of a graph as shown in Fig. 4 in the main paper, with $k-$nearest neighborhood size $k = 5$. This visualization provides a more in-depth view of how the adversarial process leads to a remapping of identities. In order to avoid visual clutter, a subset of identities and targets was chosen. As can be seen, instead of being a collapse of a facial stereotype, each hub is associated with a diverse looking set of identities, giving rise to the deep sanitization of the representation.

## 6 Qualitative Results

Figure 7 shows different reconstructions of additional CelebA identities (equal male and female) at different privacy levels. Each column is two different samples from CelebA (one male and one female), and from top to bottom, the privacy disclosure is decreasing for each. It can be noticed that visualizations from residual latent part and target latent parts confirm the sanitization visually.

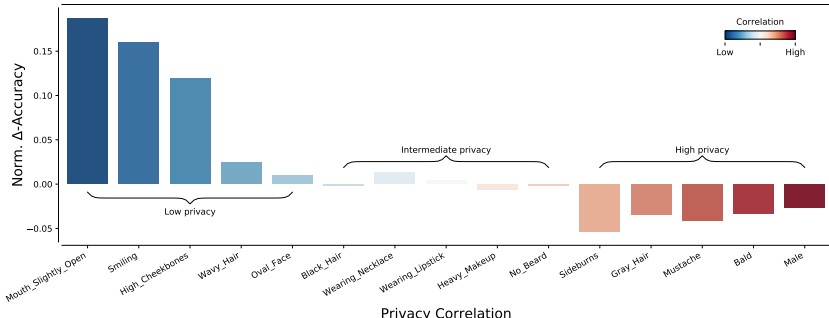

Figure 6: Attribute-level Privacy Analysis: The normalized $\Delta$-Accuracy and privacy trade-off on CelebA dataset. See Tab. 2 for detailed results.

## 7 Attribute-level Privacy Analysis

In Tab. 2 we report numerous statistics on CelebA attributes. Specifically, we analyze the attribute classification accuracy behavior w.r.t. varying level of privacy of attributes. To our knowledge, this is the first exploration of attribute-level privacy-preserving representation learning algorithms on the Celeb-A dataset. The rich correlation structure amongst all attributes makes this a challenging privacy dataset; i.e., it is difficult to achieve high accuracy for non-private attributes and low accuracy for private ones. Therefore representative subsets of the original attributes were chosen for a more in-depth analysis as indicated by the first column in Tab. 2. The subsets were chosen on the grounds of different values of correlation with IDs, reflecting different privacy levels of each and, as such, exhibiting "relatively unbiasedness" in terms of frequency within the dataset. Each of the three subsets (low, intermediate, and high privacy, resp.) consists of 5 attributes. Data statistics reported comprise normalized accuracy (w.r.t. prior) of the classifiers trained on the target representation $z_{tar}$, the $\Delta$-accuracy for each attribute corresponding to the difference of the accuracy between two classifiers — the one trained on $z_{tar}$, and the one trained on $z_{res}$. The last but one column corresponds to the correlation between the level of "privateness" and each attribute. As it can be seen in the table, the more unrelated a variable is to the identity, the higher gains in accuracy manifest. Finally, the last column corresponds to the frequency (prior) of the attribute in the dataset. For a visualization of the privacy-attribute correlation, see Fig. 6. As can be seen, the higher the correlation to privacy, the higher appears the loss in accuracy. For example, privacy-revealing attributes such as gender and facial attributes strongly correlate, therefore exhibiting a significant loss in accuracy. In contrast, attributes related to temporal facial features such as gestures feature low privacy correlation and therefore achieve strong accuracy. Conversely, the more unrelated a variable is to the identity, the higher gains in accuracy manifest.

## 8 Architectural Details

We describe the architectures of each part of our model. Table 5 shows the architectures of the VAE, i.e., the encoder and the decoder. It should be noted that the last two layers of the encoder in Tab. 3 arise from layer splitting to accommodate for partitioning target and residual representations. This is highlighted with dashed lines. Furthermore, we provide the architectures of classifiers in Tab. 6. Architectures for target and adversarial classifiers are identical.

| CIFAR-100 Krizhevsky (2009) | | | | | |
|---|---|---|---|---|---|
| Group | Attribute | Target Accuracy | Δ-Accuracy | Correlation | Prior |
| Low | Mouth_Slightly_Open | 0.92 | 0.187 | 0.222 | 0.483 |
| Low | Smiling | 0.912 | 0.16 | 0.239 | 0.482 |
| Low | High_Cheekbones | 0.861 | 0.119 | 0.267 | 0.455 |
| Low | Wavy_Hair | 0.801 | 0.0248 | 0.326 | 0.32 |
| Low | Oval_Face | 0.738 | 0.00962 | 0.332 | 0.284 |
| - | Attractive | 0.789 | -0.0144 | 0.335 | 0.513 |
| - | Pointy_Nose | 0.752 | -0.0248 | 0.351 | 0.277 |
| - | Straight_Hair | 0.791 | -0.0437 | 0.356 | 0.208 |
| - | Bags_Under_Eyes | 0.835 | 0.0151 | 0.358 | 0.205 |
| - | Brown_Hair | 0.849 | 0.00913 | 0.361 | 0.205 |
| - | Arched_Eyebrows | 0.826 | -0.006 | 0.369 | 0.267 |
| - | Wearing_Earrings | 0.865 | 0.0569 | 0.371 | 0.189 |
| - | Big_Nose | 0.814 | -0.0635 | 0.396 | 0.235 |
| - | Narrow_Eyes | 0.89 | 0.0404 | 0.402 | 0.115 |
| - | Bangs | 0.949 | 0.0819 | 0.405 | 0.152 |
| - | Bushy_Eyebrows | 0.893 | -0.0505 | 0.419 | 0.142 |
| - | Blond_Hair | 0.936 | -0.0159 | 0.429 | 0.148 |
| - | Big_Lips | 0.78 | -0.0721 | 0.434 | 0.241 |
| Intermediate | Black_Hair | 0.87 | -0.00142 | 0.370 | 0.239 |
| Intermediate | Heavy_Makeup | 0.89 | 0.0131 | 0.380 | 0.387 |
| Intermediate | Wearing_Necklace | 0.882 | 0.00404 | 0.403 | 0.123 |
| Intermediate | Wearing_Lipstick | 0.906 | -0.00617 | 0.415 | 0.472 |
| Intermediate | No_Beard | 0.935 | -0.00149 | 0.435 | 0.835 |
| - | 5_o_Clock_Shadow | 0.913 | -0.0467 | 0.435 | 0.111 |
| - | Wearing_Necktie | 0.941 | 0.0509 | 0.437 | 0.0727 |
| - | Receding_Hairline | 0.931 | -0.0212 | 0.439 | 0.0798 |
| - | Blurry | 0.956 | 0.125 | 0.443 | 0.0509 |
| - | Rosy_Cheeks | 0.95 | 0.0581 | 0.448 | 0.0657 |
| - | Eyeglasses | 0.987 | 0.0752 | 0.457 | 0.0651 |
| - | Chubby | 0.952 | -0.0341 | 0.458 | 0.0576 |
| - | Wearing_Hat | 0.982 | 0.0657 | 0.459 | 0.0485 |
| - | Double_Chin | 0.958 | -0.0421 | 0.46 | 0.0467 |
| - | Pale_Skin | 0.967 | 0.108 | 0.461 | 0.0429 |
| - | Goatee | 0.955 | -0.0446 | 0.461 | 0.0628 |
| - | Young | 0.866 | -0.0211 | 0.462 | 0.774 |
| High | Sideburns | 0.957 | -0.0531 | 0.463 | 0.0565 |
| High | Gray_Hair | 0.971 | -0.0344 | 0.471 | 0.0419 |
| High | Mustache | 0.965 | -0.0413 | 0.474 | 0.0415 |
| High | Bald | 0.984 | -0.0332 | 0.486 | 0.0224 |
| High | Male | 0.958 | -0.0262 | 0.494 | 0.417 |

Table 2: CelebA attribute-level privacy analysis. The group label indicates whether, and for which group attribute was selected for visualization of Fig. 3 in the main paper. Accuracy on the target and residual representation part, respectively. Prior is the attribute frequency bias in dataset split.

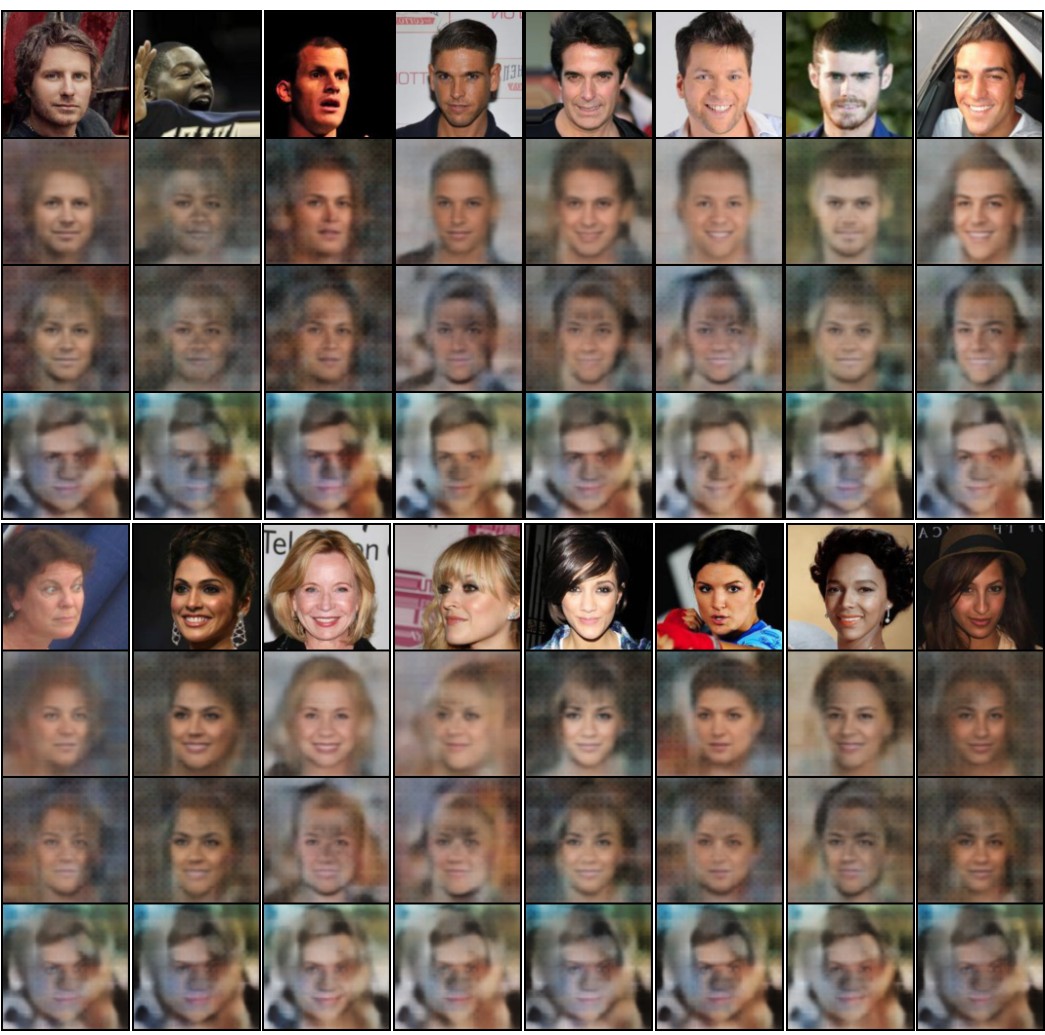

Figure 7: Visualization of CelebA data and reconstructions at different privacy levels. (From top to bottom, privacy revelation is decreasing).

| Layer | Output | Parameters |
|---|---|---|
| Input: $128 \times 128 \times 3$ | | |
| Conv-2d | $64 \times 64$ | $64 \times [3 \times 3]$, st. 2 |
| BatchNorm | | |
| LeakyReLU | negative slope: 0.01 | |
| Conv-2d | $32 \times 32$ | $128 \times [3 \times 3]$, st. 2 |
| BatchNorm | | |
| LeakyReLU | negative slope: 0.01 | |
| Conv-2d | $16 \times 16$ | $256 \times [3 \times 3]$, st. 2 |
| BatchNorm | | |
| LeakyReLU | negative slope: 0.01 | |
| Conv-2d | $8 \times 8$ | $512 \times [3 \times 3]$, st. 2 |
| BatchNorm | | |
| LeakyReLU | negative slope: 0.01 | |
| Linear | $1 \times 4096$ | |
| Linear | $1 \times 512$ | |
| Linear | $1 \times 4096$ | |
| Linear | $1 \times 512$ | |

Table 3: Encoder

| Layer | Output | Parameters |
|---|---|---|
| Input: 1024 | | |
| Linear | 32768 | |
| BatchNorm | | |
| LeakyReLU | negative slope: 0.01 | |
| DeConv-2d | $16 \times 16$ | $256 \times [3 \times 3]$, st. 2 |
| BatchNorm | | |
| LeakyReLU | negative slope: 0.01 | |
| Conv-2d | $32 \times 32$ | $128 \times [3 \times 3]$, st. 2 |
| BatchNorm | | |
| LeakyReLU | negative slope: 0.01 | |
| Conv-2d | $64 \times 64$ | $64 \times [3 \times 3]$, st. 2 |
| BatchNorm | | |
| LeakyReLU | negative slope: 0.01 | |
| Conv-2d | $128 \times 128$ | $3 \times [3 \times 3]$, st. 2 |
| Tanh | | |

Table 4: Decoder

Table 5: Architectural details of VAE components. Parameters for convolutions correspond to: number kernels $\times$[ kernel size ], and stride. Dashed lines in the encoder denote the two separate streams.

| Layer | Output size / Params |
|---|---|
| Linear | 256 |
| BatchNorm | |
| PReLU | |
| Dropout | drop-rate: 0.2 |
| Linear | 128 |
| BatchNorm | |
| PReLU | |
| Linear | #classes |

(a) Classifier on CIFAR-100

| Layer | Output size / Params |
|---|---|
| Linear | 256 |
| BatchNorm | |
| PReLU | |
| Dropout | drop-rate: 0.5 |
| Linear | 128 |
| BatchNorm | |
| PReLU | |
| Linear | #IDs or #attributes $\times$ [1] |

(b) Normal classifier on CelebA

| Layer | Output size / Params |
|---|---|
| Linear | 256 |
| BatchNorm | |
| PReLU | |
| Dropout | drop-rate: 0.5 |
| Linear | 128 |
| BatchNorm | |
| PReLU | |
| Dropout | drop-rate: 0.5 |
| Linear | 256 |
| BatchNorm | |
| PReLU | |
| Linear | #IDs |

(c) Strong ID classifier on CelebA

Table 6: Architectural details of the used classifiers

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
