# OpenReview forum: "Learning Private Representations with Focal Entropy"
_ICLR.cc/2021/Conference — Reject_

### Official Review · AnonReviewer2 · 2020-10-15

**Rating:** 5
**Confidence:** 4

**Review:**


### Summary
This paper presented a method for learning private representations. This method is based on adversarial representation learning and the main technique contribution comes from focal entropy. The experimental results show that focal entropy can improve the accuracy of the target predictor without increasing adversarial accuracy.

### Pros
1. Focal entropy is effective in reducing the information leakage of the learned representation while improves the target accuracy over the state-of-the-art.
2. This paper covers a wide range of experiments.

### Cons
1. Some technique details are not presented clearly. I have listed some of them here:
   a. How to divide the representation $z$ encoded by the encoder into two sub-representations, namely, target and residual representations.
   b. There is a high-level description of focal entropy instead of equations.
2. Some notations are very confusing. For example, in the optimization goal (Equation (2)), the adversarial loss $\phi_{tilde{S}}$ is related to the parameter $\theta_{res}$. However, in the following description, this loss is related to $\theta_{tar}$(Equation (6))

### Comments
1. The optimization objective involves many individual terms. The experiments should further provide results to show how the trade-off parameters $\beta$ controls privacy leakage and target accuracy.
2. This work is done in an adversarial training manner, can leakage reduction be achieved in a differentially private training manner, i.e., training the encoder using dp-sgd?

---

> ### Author Response · Authors · 2020-11-18
> **Rebuttal:**
>
> We thank the reviewer for constructive feedback.
>
> **1) “How to divide the representation $z$ encoded by the encoder into two sub-representations, namely, target and residual representations.”**
> The representation is divided equally, with dimensionality was chosen to be comparable to other approaches. Dividing the representation imposes no hard restriction -- as long as the latent dimensionality provides ample capacity for the downstream tasks. Furthermore, to ensure sufficient capacity and avoid degenerate solutions, we also evaluated the classification accuracy not only for the target variable on $z_{tar}$ but also of the sensitive variable on $z_{res}$.
>
> **2) “There is a high-level description of focal entropy instead of equations.”**
> To make the concept of focal entropy more clear, we added equations (9) and(10) to the manuscript, which details the recalibration of probabilities to achieve the off-centered maximum of the entropy.
>
> **3) "Some notations are very confusing. For example, in the optimization goal (Equation (1)), the adversarial loss on target is related to the parameter $\theta_{res}$. However, in the following description, this loss is related to $\theta_{tar}$ (Equation (4))"**
> We thank the reviewer for the hint. To make the manuscript clearer and avoid potential misunderstanding, we adapted the notation, added color coding, and the corresponding graphical model ( see Fig. 1).
>
> **4) “This work is done in an adversarial training manner, can leakage reduction be achieved in a differentially private training manner, i.e., training the encoder using dp-sgd?”**
> Our method is fundamentally different from DP-SGD, as we aim to learn a private *representation* instead of preserving privacy at the *parameter* level. While we do not consider a DP framework here, our method could employ differential privacy during the post-classifier training. We emphasize that the methods dealing with DP are rather orthogonal to the direction of the proposed work. So focal-entropy can be integrated seamlessly into DP frameworks, making representation learning more robust.

---

### Official Review · AnonReviewer4 · 2020-10-26
**I found the paper confusing, obscuring what might be a nice idea & experiments**

**Rating:** 4
**Confidence:** 4

**Review:**

Summary: This paper gives a method in the class of learning representations which have some information censored. In particular, the authors propose a setup where there are many “private” classes, and some classes are more similar than others – this maps (I think) onto the privacy setting, where each class is like one individual. They give a modification of an entropy loss, focal entropy, which is conducive to this type of learning, and show in experiments that this method can be successful.

I recommend reject due to a lack of clarity in the formulation, motivation, and writing of this paper. I think the idea has promise, the experiments are fine, and if correctly communicated can be a successful paper, but as it stands I found the paper pretty confusing and I’m not really sure what the method is for.

Strong points:
-	The problem as I understand it is interesting and the work is well-situated in the privacy literature.
-	Focal entropy seems like a good idea and as far as I know is novel as a loss.
-	Experiments on CIFAR and CelebA demonstrate some good behaviours from this method, mostly beating baselines
-	Experiments are mostly good, decently thorough

Weak points and Clarifications:
-	The main weak point of this paper is the exposition and clarity – I find that the problem setup is not explained particularly well. Especially as someone who is more familiar with the fair representation learning literature, I get confused when the authors refer to a sensitive attribute, or private part of the data, in this setting – it seems to be a different notion than I am used to and it is never clearly defined. If I squint I can see how it maps onto privacy nicely but I would prefer if the authors make that clear.
-	The introduction could use a rewrite – it doesn’t set up the main points of the paper particularly clearly and leaves the reader a little confused. Consider stating more clearly off the top what the problem statement is, and move much of the content of p1 to related work
-	The motivation of “highly overlapping information” is pretty imprecise. Having read the paper I can kind of see what you’re getting at but it’s not clear – I think you mean instead information hierarchy? Sub and super classes
-	As far as I can tell the exact equation for focal entropy is never actually given? Let me know if I’m wrong
-	Figure 3b – you mentioned the better tradeoff in the high accuracy domain, but not what appears to be the worse tradeoff in the low accuracy domain. Is there anything I’m missing about understanding this figure?

Other feedback:
-	P2 “the solutions mentioned earlier can only meet its practical promises …” – need a citation for this or explanation
-	P2: explain these suboptimalities more, I’m not sure exactly what you’re referring to
-	P2: you bring up a “vast number of dissimilar classes” without explaining why that’s relevant – in much work we deal with binary attributes/classes
-	Eq 4: looks a lot like standard fair representation learning, so need to explain more clearly in motivation why it’s different
-	“Although maximization of entropy is sufficient for nonprivate attributes to minimize information leakage across representation partitions, we postulate that proper sanitization must be conducted w.r.t to focus classes in a similarity-aware fashion” – this sentence is very important to the paper, and I don’t quite understand any of it. If you work hard on this sentence (why do you postulate that? What is a focus class? Why would we ever consider similarity-awareness and what is that?) you’ll go a long way to clarifying your intro
-	Top of 3.2 – need to clarify what s is
-	Eq 7 – this makes it look like Y_similar is a set of tuples but I think you mean set of elements of Y?
-	“would like the entropy peak shifted such that uniformity wrt similar classes gets dominant” – this is another idea which is not being clearly communicated right now and seems central to this work
-	The “reweighting vector” is not introduced – not clear what we will do with it. You never actually show how this is included in the entropy calculation
-	Would like to see some sort of statistic on hubs, along with these pictures

---

> ### Author Response · Authors · 2020-11-18
> **Rebuttal:**
>
> **“Problem setup is not explained well”
> "Difference to the fair RL**
> The proposed approach does not specifically consider the notion of fairness. However, as the concepts are related our method theoretically could also be extended in this regard, leading to exciting future research. A key difference to fairness approaches is that our approach deals with attributes exhibiting *different levels* of privateness. Specifically, the model does not have any prior info about each attribute's “privateness” level. It should learn a representation, which contains information about non-sensitive attributes while refraining from encoding sensitive attributes. As far as we know, no previous work in fairness literature looked at this. In contrast to that, the notion of sensitive and non-sensitive attributes is typically given a-priori in the fairness literature.  To shed more light on the correlation between privacy and individual attributes, we added the “Attribute-level Privacy Analysis” results to the appendix. It manifests that the more related an attribute is to the identity, i.e., the higher the privateness, the lower the achievable accuracy (e.g., male and mustache).
>
> **“Motivation of “highly overlapping information”**
> With overlap in information, we refer to the correlation between sensitive and non-sensitive, i.e., $S \not\perp T$. This is crucial, as this determines the degree of possible disentanglement in the adversarial setup. Whereas CelebA does not permit such clear segregation between public and private due to the inherent correlation, CIFAR-100 does permit such a separation due to the ‘artificial’ hierarchy between the sub and superclass.
>
> **“Fig 3b - trade-off curve”**
> Please, see answer *Results* to *AnonReviewer1*.
>
> **“Exact equation for focal entropy ”
> “reweighting vector is not introduced”**
> To make the concept of focal entropy and its optimization clearer, we added Equation (9) and (10) in the revised manuscript in Section 3. It explicitly presents how to obtain off-centered entropy.
>
> **Clarifying: “proper sanitization must be conducted w.r.t to focus classes”and “entropy peak shifted such that uniformity wrt similar classes gets dominant”**
> The proposed approach builds upon the idea of calibrating the distribution matching for the sensitive attribute, tailoring it to a privacy setup. This is achieved by a simple yet effective modification of Shannon entropy. Shannon-entropy has its maximum in a class-agnostic fashion at uniform distribution. Disregarding the inherent structure makes optimization susceptible to shallow sanitization (shortcuts). In contrast, focal entropy enforces group-wise equiprobability by dividing the probability mass. Hence, maximization with focal-entropy is analogous to an asymmetric entropy. Asymmetry leads to inverse skew-sensitivity due to matching an imbalanced distribution.  The high-level notion of focal entropy is further illustrated in Fig. 2. Please also see the discussion about *equilibrium* with *AnonReviewer1* for more details. We added equations (9) and (10) in Section 3 to make the focal entropy concept clearer.
>
> **Clarifying: “Explain the suboptimality”and “the solutions mentioned earlier can only meet its practical promises when the private attr. do not strongly correlate with the target attr.”**
> Standard ARL entails maximization of two likelihoods. First, an adversarial network that seeks to classify and extract sensitive information from a given representation. Second, an encoder network seeks to infer a compact data representation while preventing sensitive information leakage. This results in a zero-sum minimax game formulation, which is practically sub-optimal from the perspective of preventing information leakage. This is due to the requirement of two conditions to be fulfilled to reach optimality: First, the existence of an equilibrium of minimax game players, and second, the practical optimization ability procedures to converge to such an equilibrium. Inability to reach equilibrium in terms of likelihoods consequently entails leaking the most amount of information. In contrast, optimizing for the uniformity in distribution over the sensitive labels (maximization of entropies) provides no information to the adversary and hence is less likely to leak sensitive information. However, the optimality condition of uniform log-likelihood of the predictor is predicated on the existence of an optimal discriminator (oracle classifier) and equilibrium between the players, which in practice is unattainable. Please see *(Roy & Bodeti, 2019)* for a comprehensive discussion.
>
> **“Would like to see some sort of statistic on hubs, along with these pictures”**
> In the revised manuscript, we added the average node degree for each graph. Starting with conventional entropy and then continuing to focal entropy - the average node degree decreases monotonically with the growing size of k-NN. See the very end of section 4.2.

---

> > ### Comment · AnonReviewer4 · 2020-11-18
> > **Response**
> >
> > - Problem setup: I see that this isn't the same as fair RL but I struggle to figure out exactly what it is. Perhaps some examples in the text would help - for instance by the time that I see that ID is the sensitive attribute in CelebA I have a better understanding; I wonder what other reasonable examples there are of target attributes? This would help greatly to explain the setup. And in this case, is the ID the sensitive attribute? or are the sensitive attributes the ones which allow you to infer ID? This is also a source of confusion for me
> >
> > - Overlapping information: There is in fact fair RL work on this topic - for instance, learning representations under an equalized odds constraint (conditional independence). Your example (beard and gender) is quite unrelated to the experiments - maybe more pointed examples would help here as well.
> >
> > - Fig 3b - if only the high accuracy models matter, then why show the rest of them? If that's your criteria of evaluation I recommend being clear about that
> >
> > - Thank you for including the focal entropy equation, that is helpful.
> >
> > - Clarifying "proper sanitization ...": I'm still pretty unclear on this sentence. I understand the concept of focal entropy but I don't really follow the connection to representation learning and sensitive information.
> >
> > - this explanation on the failings of adversarial RL is good and should be expanded upon in the paper where the claim is made - as far as I know it's not something to expect readers to know
> >
> > - thank you for the hub statistics, this is useful to see

---

> > > ### Author Response · Authors · 2020-11-20
> > > **Rebuttal**
> > >
> > > We thank the reviewer for the valuable and constructive feedback.
> > >
> > > **Problem setup: I see that this isn't the same as fair RL but I struggle to figure out exactly what it is. Perhaps some examples in the text would help - for instance by the time that I see that ID is the sensitive attribute in CelebA I have a better understanding; I wonder what other reasonable examples there are of target attributes? This would help greatly to explain the setup. And in this case, is the ID the sensitive attribute? or are the sensitive attributes the ones which allow you to infer ID? This is also a source of confusion for me**
> > > Yes, sensitive attributes (S) correspond to the ID labels in the case of privacy-preserving RL, and the target attributes (T) is everything related to the desired downstream task. Examples of target attributes in the context of CelebA are 'sun-glasses,' 'gender,' 'beard,' etc. - see Table 2 in the supplementary material. The goal of ARL here is to learn a model that generates ‘sanitized’ data representations. This implies that the data representation facilitates training a post-classifier on it for a target attribute classification task. Simultaneously, it does not enable learning a classifier for sensitive attributes (i.e., ID). However, “Attribute-level Privacy Analysis” reveals that the accuracy of individual attributes that allow inferring ID is lower than those that correlate less with the ID.
> > >
> > > **Overlapping information: There is in fact fair RL work on this topic - for instance, learning representations under an equalized odds constraint (conditional independence). Your example (beard and gender) is quite unrelated to the experiments - maybe more pointed examples would help here as well**
> > > Thanks for mentioning the related works in fairness literature. We agree fair RL is a related notion to our private RL problem, so we will cite relevant works and discuss them in the final version of the paper.
> > > Examples that expose correlations between sensitive attribute and target attribute are: ‘Mahatma Gandhi’ and ‘bald,’ ‘eyeglasses,’ ‘Stevie Wonder’ and ‘sunglasses,’ ‘Queen Elizabeth II’ and ‘wearing hat,’ ‘Naomi Campbell’ and ‘high cheekbones,’ ‘attractive,’ to name a few examples. In these examples, one or more target attributes are the characteristic attributes of the person. We will adapt the paper to accommodate the points.
> > >
> > >
> > > **Fig 3b - if only the high accuracy models matter, then why show the rest of them? If that's your criteria of evaluation I recommend being clear about that - Thank you for including the focal entropy equation, that is helpful.**
> > > The ‘privacy’ and ‘utility’ both matter simultaneously in private RL, which is why in Fig. 3, we provide the performance across the entire trade-off spectrum. However, in our experiments, we emphasized more on the accuracy gain achieved in terms of utility, as it is actually where the challenge lies in the CIFAR benchmark. We will clarify this point about the evaluation in the paper. In fact, the utility is an often-neglected component in the privacy-preserving ML literature. This is one of the main reasons why lots of works on privacy can not be utilized in practice despite its theoretical merits. What makes our method superior compared to the competitors is that we managed to achieve the accuracy gain at the same or higher level of privacy.
> > >
> > >
> > > **Clarifying "proper sanitization ...": I'm still pretty unclear on this sentence. I understand the concept of focal entropy but I don't really follow the connection to representation learning and sensitive information.**
> > > Focal entropy incorporates the inter-class similarity in the entropy maximization formulation. It enforces the representation of the samples from ‘similar’ classes to be more indistinguishable from each other. This phenomenon sanitizes the representation from class information, such that (ideally) no later post-classifier can maliciously exploit it. Let us make this more concrete with a CelebA example. Given an image of ‘Yoshua Bengio,’ focal entropy would maximize the entropy w.r.t. ‘Samy Bengio’ rather than with ‘Yann LeCun.’
> > > Consequently, a hub would emerge, usurping ‘Yoshua Bengio’ and ‘Samy Bengio’’. As this hub is only sanitized w.r.t. sensitive information that allows distinguishing between them, information of target attributes remains largely untouched. This explains the comparably high utility of the proposed approach compared to alternative approaches. See the end of Section 3.2 for a comprehensive discussion and supplementary material (5.2 Adversarial Identity Mapping) for empirical evidence of it.
> > >
> > > **this explanation on the failings of adversarial RL is good and should be expanded upon in the paper where the claim is made - as far as I know it's not something to expect readers to know**
> > > We are happy that our explanation on failings of Adversarial RL made this point clear to you, so we will expand the paper accordingly in the final version of the manuscript.

---

### Official Review · AnonReviewer1 · 2020-10-28
**Blind Review**

**Rating:** 6
**Confidence:** 2

**Review:**

## Summary
- The paper tackles the problem of adversarial representation learning i.e., to learn a low-dimensional representation of the image that encodes target-relevant non-sensitive information, but not (often correlated) sensitive information.
- The approach follows a VAE-based adversarial framework, with the core novelty being the use of maximizing focal entropy  (i.e., entropy among a set of similar attributes) over sensitive attributes in the representation.
- Evaluation performed on CIFAR100 and CelebA indicates that the approach outperforms similar baselines under certain conditions.

---

## Strengths

**1. Focal entropy**
- I appreciate the insight exploited by the authors -- to encourage representations with high entropy over a small set of correlated sensitive attributes

**2. Evaluation**
- I found the evaluation quite thorough. The paper compares with many recent baselines, demonstrates performances curves and additional analysis to explain the model's behaviour.

---

## Concerns

### Major Concerns

**1. Objective**
- There are a few things unclear to me in the objective (Eq. 1-6) and would appreciate if the authors clarified them.
- (a) I don't get the motivation for the two pairs of the classifiers $(T, S)$ and adversarial $(\tilde{T}, \tilde{S})$ that tries to simultaneously minimize/maximize both the target $p(a|z)$ and sensitive attributes $p(y|z)$.
- (b) Especially, I don't understand the reasoning behind having an adversarial loss on the target attribute -- shouldn't this be maximized in all cases?
- (c) I am also confused with the sanitization term (Eq. 6). In particular of why the sensitive attribute classifier $\tilde{S}$ is a function of the target attribute classifier $\tilde{\theta}_{tar}$?
- (d) The term $\phi_{\tilde{T}}$ appears to be undefined to complement Eq. 6.
- (e) It would also be nice if the authors extended Eq. 6 (which maximizes standard entropy) with the final objective to maximize focal entropy.
- (f) Is the VAE term (Eq. 5) necessary (esp. reconstruction)? After all, this appears as a secondary objective opposed to the primary objective of learning a representation $z$ which minimizes information leakage.

**2. Writing - Sec. 3**
- My issues in understanding can be partly attributed to the writing in Sec. 3, which I found difficult to follow for a few reasons.
- (a) Some terms seem to be overloaded e.g., two notations for encoder $q(x; \theta_E)$ and $E(x; \theta_E)$. I was also thrown off by many ways the components of the model/objective are conveyed: Losses $\phi_T$, parameters $\theta_T$, player $T$, adv. player $\hat{T}$, etc. I recommend finetuning the second paragraph on page 4.
- (b) While the architecture in Fig. 1 is designed to accompany the text, I find it somewhat incomplete and unclear. For one, there are six players/blocks in the proposed architecture. However, only the encoder and decoder are shown in the figure. Furthermore, while there are five loss terms, only four of them are shown in the figure. I am also not sure if the blue/red backgrounds in the residual and target streams code something in particular. Perhaps one solution is to split the figure into two: one for the architecture and another for information flow?

**3. Results**
- While the focal entropy makes sense to me (i.e., by increasing entropy over a small set of similar classes), I wonder if the results confidently back up that is it indeed better than related baselines.
- (a) The improvements seem somewhat marginal e.g., an improvement of 2% accuracy (Table 1) over prior work on both CIFAR100 and CelebA.
- (b) But what I find more revealing of the performance is the trade-off curve in Fig. 3b (thanks for the presenting this!). It appears that the proposed focal entropy approach (blue curve) outperforms Kernel-SARL (red curve) only for in a small operating range of high target accuracy. Overall it appears that MaxEnt-ARL and Kernel-SRL offers better trade-offs.


### Minor Concerns

**4. Decoder/Reconstruction**
- Since the task is to partly perform reconstruction as well, I find missing evaluation on how good the reconstructed samples are.

**5. Same equilibrium issues as before?**
- The introduction (second paragraph, p2) remarks that the adversarial min-max formulation of the problem has an issue that there is significant leakage if the optimization does not reach equilibrium.
- Wouldn't this be an issue in the proposed work as well?

### Nitpicks

**6. Some nitpicks**
- Please label the axes in Fig. 2.
- Not sure what this sentence means "increases the uncertainty in a more organic fashion" - please rephrase
- Typo in citation "Radovanovi263 et al. (2010)"

---

> ### Author Response · Authors · 2020-11-18
> **Rebuttal:**
>
> **“(a) Motivation for two pairs of the classifiers”
> “(b) Reasoning behind adv. loss on target”**
> The non-adversarial predictors enforce the utility and the presence of the associated information in the representation (target information in the $z_{tar}$ and sensitive in $z_{res}$, the adversaries inhibit undesirable information leakage across the representation partitions. Thus are responsible for disentanglement and sanitization. Moreover, as the adversaries are learned during training, they act as surrogates for oracle post-classifiers.
>
> **(c) “Why is $\tilde{S}$ a function of $\tilde{\theta}_{tar}$?”
> (d) “Adv. predictor is not defined”
> (e) "Extend Eq. 5”**
> We significantly modified the method section and incorporated the suggested hints. Specifically, we adapted the notation towards simpler terms and fixed typos that confused. In terms of the sanitization terms, we added Equation (9) and (10) to make the focal entropy concept and its optimization clearer.
>
> **(f)  VAE**
> Integration of VAE into the proposed approach was done to *i)* improve representation learning accompanied by disentanglement, and *ii)* provide interpretability.
> *i)* It has been shown that integration VAEs into representation learning in combination with supervision boosts the performance of downstream tasks (Le et al. (2018); Gyawali et al. (2019)). We similarly observed higher predictor acc. Additionally, VAEs are known to promote disentanglement, which can be attributed to the encoder KL-term via independent priors (Burgess et al., (2018)). In our scenario, this serves the objective of separating sensitive and non-sensitive attributes into separate subspaces.
> *ii)*  Interpretability is crucial yet often overlooked in privacy literature. Integrating the VAE into the minimax game facilitates understanding the sanitization in a more human interpretable fashion.
>
> **"increases the uncertainty in a more organic fashion"**
> This refers to the removal of sensitive information in a semantic-aware and systematic fashion, leveraging the inherent class structure. Given the example of CIFAR, we want a 'whale' to be more confused with another type of fish, rather than, e.g., with the class ‘vehicles’. Doing so helps eliminate ‘fine-grained’ features, giving rise to the guided sanitization.
>
> **Results**
> Given a correlation between the target and sensitive attributes, the minimax nature of the ARL entails a compromise between privacy and utility.  The proposed approach accommodates this trade-off in a principled fashion, as changing the *'' focus''* and the size of it (i.e., nearest neighbors) directly affects the shape of the trade-off response curve (see Figure 2). In this respect, *MaxEnt-ARL* (Roy & Bodeti, 2019)} which imposes uniformity in a class agnostic fashion is a special case of our proposed focal entropy. In the trade-off curve, focal-entropy performs significantly better than *MaxEnt-ARL* in the relevant operating range (i.e., high target acc.).
> We also outperform *Kernel-SARL* sharply in the ‘linear’ case. In the ‘kernelized’ case, we gained higher performance in the targeted utility domain as well (achieving 2% accuracy improvement at an identical privacy rate is extremely challenging ). The ‘kernelized’ case, however, cannot be directly optimized in an end-to-end fashion, so it is not directly applicable to DNN-based solutions(as admitted by the authors). Furthermore, the performance reported by both methods is obtained as non-dominated solutions and may not be directly comparable.
>
> **“leakage of information when not achieving equilibrium”**
> Employing *ML-ARL* is subject to information leakage due to optimization of log-likelihood for sanitization. In contrast, the theoretical optimality of MaxEnt-ARL for sanitization is predicated on the existence of an optimal discriminator (oracle classifier) and minimax equilibrium. Despite the theoretical optimality, the objective of the latter is unattainable in practice. Our proposed Focal-entropy method goes towards maximization of entropy by integrating MaxEnt-ARL as a special case. This is achieved trading in optimality with quasi-optimality by means of ‘approximating’ the optimization objective. Such approximation is simply achieved by enforcing two separate uniform distributions for the ‘similar’ and ‘dissimilar’ classes (see revised manuscript for a discussion).
>
> **Evaluation on reconstructed samples**
> The purpose of reconstruction is to improve representation learning and interpretability. As the reconstruction quality was a minor point for us, we just added some example reconstructions to the supplementary material. However, we consider reporting the Inception Score of the reconstruction of our method (with the final version of this manuscript) for the sake of future comparison.
>
> **“Some terms seem to be overloaded”**
> We adapted and, in particular, streamlined the notation to avoid misunderstanding and clutter.

---

### Official Review · AnonReviewer3 · 2020-10-29

**Rating:** 6
**Confidence:** 3

**Review:**

########################################################################## Summary: The paper studies how to learn private representations that only captures the non-sensitive attributes of the dataset. They propose an adversarial representation learning method that employs VAEs. Specifically, the architecture in the VAE contains 6 players with 2 as adversarial classifiers. They introduce focal entropy as the objective function instead of entropy for adversarial classifiers to achieve deep sanitization. They empirically evaluate the method by reporting the target task accuracy and attribute inference accuracy on two datasets.

########################################################################## Reasons for score: I like the paper largely. It provides a nice practical method for sanitization, although it is hard to give theoretical privacy guarantees of VAEs. The method has shown to be a good defense for individual attribute inference attacks. My main concern is the experimental results are only on two datasets with one task/sensitive attribute setting. As an empirical/methodology paper, I would expect more empirical results.

Also, I have a question regarding the sanitization section. It's not clear if focal entropy should be used in both $\tilde{T}$ and $\tilde{S}$. The paper mentioned "training of $\tilde{T}$ leverages a modification of entropy ..." However, I think it should be $\tilde{S}$ or both. It seems that Equation (6) should be negative KL divergence since we want to force the distribution to be close to the uniform distribution.

########################################################################## Minor Comments:

1. In Equation (6), the bracket should be after $\theta$.

2. I don't think $\pi$ is clearly mentioned before Equation (9). The notation should be more precise.

 3. I'm guessing it can also protect the dataset-level (proprietary) attribute inference. I'm interested in seeing the results, but it is not required for this submission.

---

> ### Author Response · Authors · 2020-11-17
> **Rebuttal:**
>
> We thank the reviewer for the valuable and constructive feedback. We incorporated the feedback of the review in the revised manuscript.
>
> **1) “My main concern is the experimental results are only on two datasets [...]"**
> We perform an extensive set of experiments on two standard benchmarks. Please also see supplementary material. As the first testbed, we adopt the “simulated” privacy problem proposed by Roy & Boddeti (2019) designed on the CIFAR-100 dataset. We picked this benchmark as it captures a “perfect” separation between target and sensitive information, i.e., $S \perp T$. We compare with state of the art adversarial and entropy-based representation learning methods for privacy on this benchmark. Second, the CelebA dataset where no clear segregation between target and residual spaces is possible, i.e., $S \not\perp T$. The “in-the-wild” nature of face images offers a richer testbed for our method as both identities, and contingent factors are significant sources of variation. We compare with many recent baselines on this challenging benchmark, demonstrate performance curves, and additional analysis to explain the model's behavior (as pointed out by other reviewers too). We agree that having more “large-scale” datasets for the evaluation would be better to substantiate the claim of the empirical effectiveness of the proposed method. However, such benchmarks are still pretty limited, which can largely be attributed to the nature of the topic -- privacy. We expect more of such datasets to emerge as the topic lately has enjoyed increased attention in the community. Nevertheless, we are convinced that the results with the proposed approach would show the same pattern on other datasets.
> To the best of our knowledge, our paper is the first work that proposes to taking the class similarity into account for the entropy of an adversary.
>
> **2) “I'm guessing it can also protect the dataset-level (proprietary) attribute inference. I'm interested in seeing the results [...]”**
> We thank the reviewer for the suggestion and agree on this point. In this regard, we added an analysis of the correlation between privacy and individual attributes on CelebA.  In line with the reviewer's expectation, our attribute-level analysis shows that our method achieves relatively lower target accuracy for dataset-level proprietary attributes compared to the more generic attributes. See the “Attribute-level Privacy Analysis” section in the supplementary material.
>
> **3) “It's not clear if focal entropy should be used in both $\tilde{T}$ and $\tilde{S}$. The paper mentioned "training of  $\tilde{T}$  leverages a modification of entropy”**
> Whereas we want the target representation to be sanitized w.r.t. sensitive attributes, we “only” enforce disentanglement w.r.t. target attributes on the residual representation. As the target attributes on the residual representation are of no privacy relevance, we leverage only conventional entropy Equation (5) to enforce disentanglement. Whereas for sanitization of the target representation, we leverage focal-entropy Equation (9),(10).
>
> **4) “I don't think π is clearly mentioned before Equation (9). The notation should be more precise.”**
> To make the focal entropy concept and its optimization clearer, we added Equations (9) and (10) in the revised manuscript. As requested, we further elaborate on the transformation of probabilities and the final objective to maximize focal entropy.

---

### Decision · Program_Chairs · 2021-01-07
**Final Decision**

**Decision:**

Reject

**Comment:**

This paper focuses on a notion of privacy in learning representations.

One of the primary concerns of the reviewers was clarity of the writing and results. Numerous concerns are mentioned in the reviews, and also more engagement with the fairness literature was desired. One reviewer felt that some of the claims in the paper were unsubstantiated, for example: understanding the sanitization process in a human-understandable visual way", "integration of a notion of interpretability". It was felt that the changes required were more than could be expected for a camera ready version. The authors are recommended to revise the paper with a particular eye for clarity to a new reader.

The notion and measurement of privacy was also considered to be somewhat shaky. It is understood that the nature of privacy considered in this paper is different from differential privacy. That said, the latter is a rigorous definition, and the one in this paper seems to be rather empirical in nature. There are no formal guarantees in terms of privacy preservation, and it is not clear whether the representations could leak information when evaluated with a different network. As privacy is a mission-critical property, some justification of why the heuristic measurement of privacy is acceptable.

As a side note, the authors should consider using the \citep command for parenthetical citations in the text.